# Different Recognition of Protein Features Depending on Deep Learning Models: A Case Study of Aromatic Decarboxylase UbiD

**DOI:** 10.3390/biology12060795

**Published:** 2023-05-31

**Authors:** Naoki Watanabe, Yuki Kuriya, Masahiro Murata, Masaki Yamamoto, Masayuki Shimizu, Michihiro Araki

**Affiliations:** 1Artificial Intelligence Center for Health and Biomedical Research, National Institutes of Biomedical Innovation, Health and Nutrition, 3-17 Senrioka-shinmachi, Settsu 566-0002, Japan; n.watanabe@nibiohn.go.jp (N.W.); kuriya@nibiohn.go.jp (Y.K.); m.yamamoto@nibiohn.go.jp (M.Y.); 2Graduate School of Science, Technology and Innovation, Kobe University, 1-1 Rokkodai, Nada-Ku, Kobe 657-8501, Japan; murata.masahiro.8r@people.kobe-u.ac.jp; 3Bacchus Bio Innovation Co., Ltd., 6-3-7 Minatojima minami-machi, Kobe 650-0047, Japan; m_shimizu@b2i.co.jp; 4Graduate School of Medicine, Kyoto University, 54 Shogoin-Kawahara-cho, Sakyo-ku, Kyoto 606-8507, Japan; 5National Cerebral and Cardiovascular Center, 6-1 Kishibe-Shinmachi, Suita 564-8565, Japan

**Keywords:** deep learning, protein feature, feature extraction, explainable artificial intelligence, integrated gradients, protein annotation

## Abstract

**Simple Summary:**

Various protein sequences are registered in biological databases, and hundreds of the sequences have recently been sequenced by way of next-generation sequencing, and then the number of sequences with unknown functions is explosively increasing. To efficiently determine the annotations, new feature extraction of protein sequences that is different from existing knowledge is required. Deep learning can extract various features based on training data. Many studies have reported deep learning models with high accuracy for predicting protein annotations; however, in the reports, which amino acid sites in protein are important for the prediction of the annotations have not been discussed among multiple deep learning models. Here, 3 deep learning models for the prediction of the proteins included in a protein family were analyzed using an explainable artificial intelligence method to explore important protein features. The models regarded different sites as important for each model, and all models also recognize different amino acids from the secondary structure, conserved regions and active sites as important features. These results suggest that the models can interpret protein sequences through different perspectives from existing knowledge.

**Abstract:**

The number of unannotated protein sequences is explosively increasing due to genome sequence technology. A more comprehensive understanding of protein functions for protein annotation requires the discovery of new features that cannot be captured from conventional methods. Deep learning can extract important features from input data and predict protein functions based on the features. Here, protein feature vectors generated by 3 deep learning models are analyzed using Integrated Gradients to explore important features of amino acid sites. As a case study, prediction and feature extraction models for UbiD enzymes were built using these models. The important amino acid residues extracted from the models were different from secondary structures, conserved regions and active sites of known UbiD information. Interestingly, the different amino acid residues within UbiD sequences were regarded as important factors depending on the type of models and sequences. The Transformer models focused on more specific regions than the other models. These results suggest that each deep learning model understands protein features with different aspects from existing knowledge and has the potential to discover new laws of protein functions. This study will help to extract new protein features for the other protein annotations.

## 1. Introduction

Protein sequence information is registered in various biological databases [1,2]. Various protein sequences are being sequenced by next-generation sequencing technology [3]. The number of unannotated sequences registered in the databases is explosively increasing, such as putative proteins, hypothetical proteins, and uncharacterized proteins. Therefore, in addition to efficiently assigning the annotations for a large number of proteins, extracting new protein features that differ from existing knowledge is required.

Deep learning automatically can learn and extract various features of input data, and the higher the model performances are, the more valid the training data are. Therefore, the utilization of deep learning is expected to discover important features and classify data based on the features [4]. Several studies have reported deep learning models for predicting protein functions [5,6], protein structures [7,8,9,10,11,12], multi-domain protein structures [13,14], protein subcellular localization [15,16], enzyme commission numbers [6,17,18,19], and products in organic synthesis [20,21]. Each model has been evaluated and compared using multiple performance evaluation parameters in machine learning tasks. Although the evaluation of model prediction accuracy is important, these studies have not sufficiently discussed which features of input data influence prediction accuracy and have not evaluated the detailed difference of the results among multiple models.

Most of the deep learning models cannot interpret prediction results without the other methods. However, deep learning has the potential to recognize extensive and new protein features that are different from existing knowledge, such as secondary structures, conserved residues, ligand binding sites, and active sites, because the models achieve more adequate prediction accuracy than previous machine learning. Model interpretability helps to know how the model reaches the results and to quantify prediction reliability [22,23]. Several studies have recently reported integrated gradients (IG) and Shapley additive explanations (SHAP), included in explainable artificial intelligence methods [24,25], to interpret prediction models and to explore important features for prediction [6,26,27]. However, the previous reports using integrated gradients have not discussed the exploration of important features and the difference of the features among multiple deep learning models.

Here, several deep learning models derived from enzyme sequences were developed to extract protein features for each amino acid residue and then to explore the validity of the features in comparison to previously reported information. As a case study, UbiD enzymes, one of the decarboxylases which biosynthesize various aromatic compounds [28,29,30,31,32,33,34,35], were used. To extract new UbiD features, prediction and feature extraction models for UbiD were built using convolutional neural network (CNN), CNN-based autoencoder (CNN-AE), and Transformer [36,37,38]. The important protein features between these models were explored by analyzing prediction scores and feature vectors derived from the models using clustering and IG (Figure 1). As a result, UbiD features could also be extracted from the different residues from the existing knowledge by these models, and the features were varied for each model and sequence, and only the Transformer model characterized a few amino acid residues as important UbiD features. The results indicate that each deep learning model extracts different protein features from each amino acid and recognizes each sequence as different. In short, the analysis of protein features using multiple explainable deep learning is required to more deeply understand proteins.

## 2. Materials and Methods

### 2.1. Dataset Construction

#### 2.1.1. Positive Data

25,294 UbiD sequences were collected as positive data from National Center for Biotechnology Information (NCBI) Protein database [1] on 31 July 2019 by searching UbiD as a keyword. The enzyme sequences that were duplicated or included non-canonical amino acids were removed. The length of amino acid residues was limited from 400 to 700 because the length of 25,135 UbiD sequences was in the range. The sequences were clustered at 95 % identity using CD-HIT [39] to remove sequence redundancy and then were split into training, validation, and test data based on the number of sequences included in each cluster.

The 3 sequences were randomly extracted from all sequences included in a cluster and were split into each data when the number of sequences was 3 or more. When the number was 2, one sequence was added to training data, and the other was added to either validation or test data. In the rest of the cases, the sequences collected from all clusters, in which the number of sequences was 1, were randomly split into training, validation, and test data, at an approximate ratio of 8:1:1.

#### 2.1.2. Negative Data

All protein sequences registered in Swiss-Prot [2] were collected as negative data on 26 April 2019. The negative data was the protein sequences except for UbiD. Some sequences were also removed in the same way as positive data construction. The highly similar negative sequences to positive sequences were omitted at 1.0 × 10^−10^ E-value using BLAST+ 2.7.1 [40]. The rest of the sequences were clustered at 90% identity using CD-HIT, and then only a single enzyme sequence from each cluster was included.

Artificial negative data were built to prevent deep learning models from judging as positive using only a few amino acids in a specific position. In total, 400 sequences whose same dipeptide amino acids continue were generated (AA…AA, AC…AC, …). The length of the upper negative data and the artificial sequence was randomly determined from 400 to 700, such as the construction of positive data. All negative data were randomly split into training, validation, and test data, at an approximate ratio of 8:1:1. Total amounts of positive and negative data are shown in Table 1.

### 2.2. Model Construction and Evaluation

CNN model, which predicts whether or not input protein sequences are target enzymes, was built. CNN-AE model to output feature vectors derived from enzyme sequences was built. The autoencoder model transformed input protein sequences to low dimensional feature vectors and outputted similar sequences to input data. Finally, Transformer model was built for prediction of target enzymes, such as CNN model, and for output of feature vectors, such as CNN-AE model (Figure 1A). The prediction scores and feature vectors outputted from the models were evaluated using clustering and IG (Figure 1B,C).

The architecture of CNN prediction and CNN-AE feature extraction models are shown in Appendix A. The 3 hidden layers were used in CNN model, and self-attention was inserted next to the second hidden layer. The 5 hidden layers were used in encoder and decoder of CNN-AE, respectively, and 200 dimensional feature vectors were outputted. Self-attention was inserted next to the first hidden layer of the encoder and the fourth layer of the decoder. One-hot matrices transformed from amino acid sequences were inputted to both models. The sequences whose number of amino acids was less than 700 were transformed to matrices using zero-padding. CNN-AE model was built using only positive data.

Transformer model, which predicted target enzymes and extracted features derived from sequence information, was built using the encoder of Transformer [38], as shown in Appendix A. Enzyme sequences were transformed to the tokens using 3-gram model. The special tokens (<CLS>, <EOS>) were used at the beginning and end of each token. <pad> tokens were added up to 700 tokens for the sequences whose number of amino acids was less than 700. In total, 64 dimensional vectors in Extract layer (Appendix A) were used to analyze feature vectors. A binary cross-entropy loss function was used to train all models.

CNN and Transformer models were trained using several batches, including only positive or only negative sequences to prevent overfitting due to imbalanced data. The positive and negative batches were separately built using random sampling without replacement, and then the models were learned for each batch in turn. If the number of the sequences that could be extracted was less than that of batch size in batch construction, the following batches were rebuilt using the first data. CNN model was trained until 4000 steps, while Transformer model was trained until 1000 steps.

CNN and Transformer models were evaluated using test data. Accuracy (ACC), AUC, F_1_ score, and Matthews correlation coefficient (MCC) were used to evaluate the prediction models. CNN-AE model was evaluated using Match rate between input sequences and output sequences, given by the following:(1)$$\mathrm{Match}\;\mathrm{rate}=\frac{Matched\;number\;of\;amino\;acids\;in\;sequence}{Sequence\;length},$$

The CNN-AE model using the epochs, where the number of sequences with Match rates 0.9 and more, and the average of Match rates were highest, were used in the following analysis. The CNN and Transformer models using the epochs, where all evaluation parameters were highest, were used. All models were built by Tensorflow version 2.1.0 [41].

### 2.3. Case Study

UbiD enzymes were used to explore important enzyme feature vectors between the deep learning models. The enzymes catalyze the decarboxylation reactions included in the ubiquinone biosynthesis pathway, which were identified in *Escherichia coli* for the first time [28,29,42]. Usual UbiD enzymes act on para-hydroxybenzoic acid-type substrates, while the other UbiD family enzymes catalyze the reversible reactions to synthesize various aromatic compounds such as protocatechuic acid and vanillic acid [30,31,32,33,34,35]. Therefore, the analysis of UbiD family enzyme features is expected to expand the diversity of aromatic compounds, which can be biosynthesized using engineered microbes. The *E. coli* UbiD secondary structures, conserved residues, ligand binding, and active sites [29,42] are shown in Appendix A. N175 and E241 residues of *E. coli* UbiD are Mn^2+^ binding sites, I178 to R180 residues, R192 to L194, R197 residues, and G198 residue are prenylated flavin mononucleotide binding sites, and D290 residue is an active site.

### 2.4. Clustering and Integrated Gradients Analyses

The feature vectors of positive data were extracted from CNN-AE and Transformer models and were clustered by k-means algorithm. A single sequence from each cluster, whose feature vector was closest to the cluster centroid, was selected as representative sequence. Then, the representative sequences in all clusters were analyzed using IG. Moreover, these sequences were compared to the UbiD enzyme of *E. coli* (*E. coli* UbiD) registered in Swiss-Prot (sp|P0AAB4|UBID_ECOLI) to compare the clustering method based on deep learning models to sequence similarity method. In the evaluation, the distances of feature vectors and bitscore of BLASTp were calculated for *E. coli* UbiD and each representative sequence.

Integrated gradients algorithm [24] is used to evaluate the important variables that machine learning models contribute to determining the prediction results. Therefore, in this study, the algorithm was applied to explore where region of the amino acid sequence each deep learning model grasped as important UbiD features in prediction, which are similar to the secondary structure and the important functional sites in the known annotations. The features based on CNN and Transformer models were extracted by IG analysis because these models were binary classification models that can find UbiD sequences from input proteins. On the other hand, the CNN-AE-based UbiD features were obtained from hidden layers in which UbiD features were included. Absolute values of IG were calculated for each amino acid residue in *E. coli* UbiD and the representative sequences using Tensorflow (Figure 1C), and the amino acid residues with high IG values were regarded as important features for the predictions. The IG values of output scores for input sequences were calculated in CNN model, while the IG values of feature vectors for input sequences were calculated in CNN-AE model. In Transformer model, both IG values were calculated. The IG values, multiple sequence alignments between *E. coli* UbiD and the representative sequences, and secondary structures of *E. coli* UbiD were visualized using Jalview 2.11.1.4 [43]. Multiple alignment sequences were built using MAFFT version 7 [44]. In this study, Xeon E5-2609 v4 1.7 GHz, memory 32 GB (Intel, Santa Clara, CA, USA), NVIDIA Quadro GP100 16 GB × 2 (Nvidia Corporation, Santa Clara, CA, USA) running CentOS version 7.4 was used.

## 3. Results

### 3.1. Model Training and Evaluation

The loss function curves for training and validation in CNN and CNN-AE models are shown in Appendix A, respectively. The loss values for training were almost the same as the loss values for validation for both models. The matching loss values suggest that the models do not tend to overfit. Test results of the CNN model were calculated for each epoch (Appendix A). The optimized CNN model was built using 4000 epochs, where 4 evaluation parameters were the highest. On the other hand, the CNN-AE model was evaluated using the Match rate. The CNN-AE model in the 2000 epochs, whose number of sequences with Match rates 0.9 and over and the average of Match rates were highest, was selected as the optimized model (Appendix A).

The loss function curve for validation in the Transformer model decreased with matching the curve for training (Appendix A), indicating that overfitting does not occur. Test results of the Transformer model are shown in Appendix A. The Transformer model was predicted with high accuracy in all epochs, and the test results were best in epoch 1000 according to all evaluation parameters. The model in the epochs was used in the following analysis.

### 3.2. Model Interpretation

UbiD feature vectors derived from CNN-AE and Transformer models were separated into 7 clusters using the k-means algorithm. From each cluster, a single UbiD sequence whose feature vector was closest to its cluster centroid was selected as the representative sequence (Appendix A). Appendix A shows the results of feature vector distance and BLASTp bitscore between *E. coli* UbiD and each representative sequence. The higher the bitscore was, the more similar the sequence was to *E. coli* UbiD. However, feature vector distance did not seem to relate to bitscore (Appendix A).

To explore what features of UbiD sequences the deep learning models learned, these models were analyzed using IG. The IG values of *E. coli* UbiD sequence for CNN, CNN-AE, and Transformer models were calculated for all amino acid residues, as shown in Figure 2 [45,46], and the IG values of all representative UbiD sequences were calculated (Appendix A). The IG values of all representative UbiD sequences were compared to secondary structural information, conserved residues, ligand binding and active sites of *E. coli* UbiD [29,42], and the IG values of *E. coli* UbiD. Appendix A shows the predicted structures by ESMFold [12] and the residues with higher IG values of *E. coli* UbiD and 3 representative sequences.

The IG values of the conserved V29 residue, the conserved P357 and P423 residues in α-helix and conserved P216 residue in β-strand were higher for conserved the V29 residue, the conserved P357 and P423 residues in α-helix, and conserved P216 residue in β-strand in *E. coli* UbiD using CNN model. On the other hand, the P48, P61, and P152 residues, which are not known as the regions of secondary structures and the conserved residues, exhibited important features for the prediction. The IG values of the same residues were not necessarily high for *E. coli* UbiD and each representative sequence, although the results of more than half of the representative sequences were similar for the P48, P61, and P357 residues. Proline residues tended to be high IG values for most of the *E. coli* UbiD and representative sequences in only the CNN model.

CNN-AE model regarded more amino acid residues of *E. coli* UbiD as important factors than other models. The IG values of conserved P234 residue, the similar (semi-conserved) Q132 and R380 residues in the α-helix, and the similar I134 and L183 residues in β-strand were high in the sequence. Moreover, the M382 residue included in the α-helix and L429 and L63 residues included in the β-strand were regarded as important amino acids. CNN-AE model also identified P61, M4, and K5 residues, which were not included in secondary structures and were not conserved. More kinds of amino acids with high IG values appeared for *E. coli* UbiD and each representative sequence. The IG values of the active sites with the substrates and binding sites with prenylated flavin mononucleotide and Mn^2+^ by *E. coli* UbiD were not high in all representative sequences using CNN and CNN-AE models.

The number of residues with high IG values for *E. coli* UbiD using the Transformer model was so smaller than those in CNN-type models. The IG results derived from prediction scores were almost the same as the results derived from feature vectors. The conserved E285 residue included in β-strand, the conserved G286 and P287 residues, and E278, Q279, and G280 residues included in β-strand exhibited high IG values. The amino acid region between 278 and 280 residues was the highest value. The high IG residues of *E. coli* UbiD were so different from those of the representative sequences in comparison to CNN-type models. Moreover, the Transformer model regarded the different consecutive amino acid residues for each representative sequence as important amino acids.

In the Transformer model, the IG values of the E285 to P287 conserved residues of *E. coli* UbiD were high, and the E285 was a putative active site in *Pseudomonas aeruginosa* [29,42]. EJZ42036.1 and WP_058074034.1 results showed the same tendency. Moreover, in AKC32612.1, the IG values of Y242 adjacent to the Mn^2+^ binding site, the 241E, were almost 1. The transformer model tended to extract UbiD features from the other residues except for annotated functional residue in the other representative sequences. In all models, the correlation coefficients between the IG values of each residue of all UbiD sequences and sequence conservation [42] were almost 0 (Appendix A).

## 4. Discussion

Functional annotations for protein sequences are required to understand cell functions and to search novel enzymes for target compound productions. However, annotating sequence functions is insufficient due to the increase in the number of unannotated proteins. Therefore, in this study, comprehensive features for accurately annotating enzyme sequences were explored by analyzing enzyme feature vectors derived from various deep-learning methods and IG values of each amino acid residue.

CNN, CNN-AE, and Transformer models for UbiD enzyme prediction and feature extraction were built and evaluated using multiple evaluation parameters. The validation results indicate that all current models do not occur overfitting because the loss values decreased as the training proceeds. CNN model was improved by increasing the number of training steps according to all parameter values, then the CNN model in the last 4000 steps was used. Then, the Transformer model predicted the enzymes with high accuracy, and test results showed constant prediction accuracy in all epochs. The Transformer model in 1000 steps where all parameter values were highest was selected, although the model in lower steps exhibited sufficient accuracy and seemed to be optimized. Moreover, the CNN-AE model in 2000 epochs generated output sequences with 0.9 and more Match rates, which were the almost same as input UbiD sequences, and therefore the model can learn sufficient UbiD sequence features.

To analyze the enzyme features derived from each model, the feature vectors built from the hidden layer of CNN-AE and Transformer model were clustered using the k-means algorithm, and 7 representative UbiD sequences were selected by each model. The distances between *E. coli* UbiD and the representative sequence feature vectors did not seem to relate to BLAST bitscores based on sequence identity. The results indicate that CNN-AE and Transformer models grasp different features from the conventional method. Moreover, the Transformer model enables us to deeply understand slight differences in each sequence because the distances derived from the Transformer model varied depending on the combinations of 2 sequences among UbiD sequences than the CNN-AE model (Appendix A).

Next, what features and amino acid residues of UbiD sequences were regarded as important for each model were explored using IG. The UbiD amino acid residues regarded as important features were different for each model. The important residues consisted of not only the secondary structures, the conserved regions, the ligand binding and the active sites but also the other regions. CNN-based models did not extract the features from active and cofactor binding sites. The results suggest that the models learn the new important enzyme feature information, which is not included in the protein database [1,2,47] and previously annotated information. Moreover, the CNN model identified the same kind of amino acid as important according to the results of the most of UbiD sequences, while the CNN-AE model showed the high IG values of the extensive residues for UbiD sequences. This is because the autoencoder model learns to ensure that the outputs match the input enzyme sequences.

The results using the Transformer model were surprisingly quite different from those of CNN-type models. The residues, which UbiD features were extracted from, were different depending on the sequences, and the residues were not necessarily important functional sites. The number of the important residues for each UbiD sequence was much smaller, although the most of residues were included in secondary structure and conserved amino acids. Therefore, the Transformer model focuses on more specific residues than CNN-type models and can extract more different enzyme features from the existing annotations for each sequence. Moreover, the Transformer model results that the different amino acid regions for each sequence showed high IG values are consistent with the results that the variance of distances between the 2 UbiD sequences was larger in comparison to CNN-type models. According to the results of the correlations between IG values and UbiD conservation, all deep learning models also extract the features that are different from the important conserved residues. In the future, the optimizations of the training datasets, especially for negative data and model structures of each model are also required for building more accurate models and extracting more higher quality features from protein sequences, such as ablation study [48,49]. Moreover, to apply the analysis to determine annotations for protein sequences, more extensive tests using the other protein families are needed because the amino acids with important features are determined depending on the models and sequences.

## 5. Conclusions

In this study, deep learning models were built using specific enzyme sequences included in one of the protein families, and the feature vectors derived from the models were analyzed using IG. As a result, the models regarded not only the amino acid residues included in not only the secondary structures, the conserved regions, the ligand binding and the active sites but also the other regions as important features. Therefore, the analysis can grasp multiple enzyme features that are different from previously reported information. Moreover, these models extracted different features from the sequences for each model and recognized each sequence with different features, even for similar sequences. These results show that building and evaluating models using multiple deep learning methods are more important to extract various protein features, which will be the basis of new knowledge because the recognitions of protein features are more different among each method. This method will help to interpret protein sequences through different perspectives from existing knowledge and to discover new features and motifs for unannotated protein sequences.

## Figures and Tables

**Figure 1 biology-12-00795-f001:**
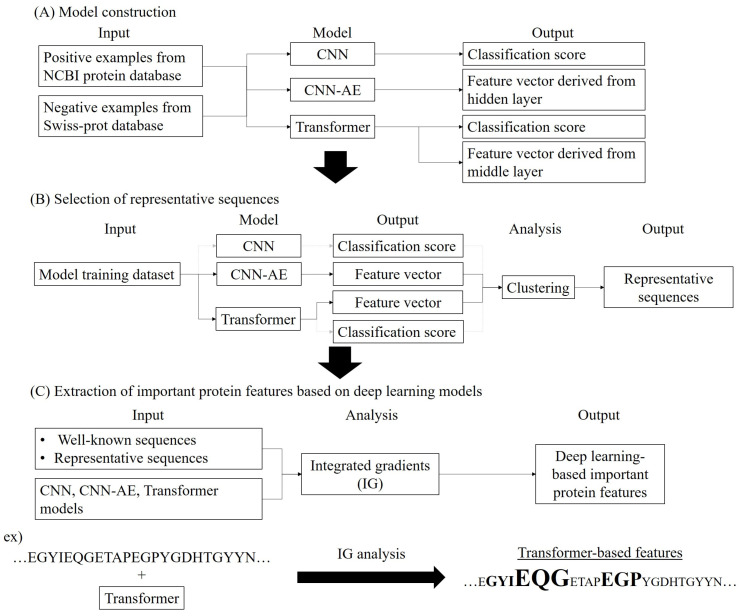
Workflow of methodology. (**A**) Model construction and model information, (**B**) Selection of representative sequences using deep learning and clustering, (**C**) Extraction of important protein features based on deep learning using Integrated gradients.

**Figure 2 biology-12-00795-f002:**
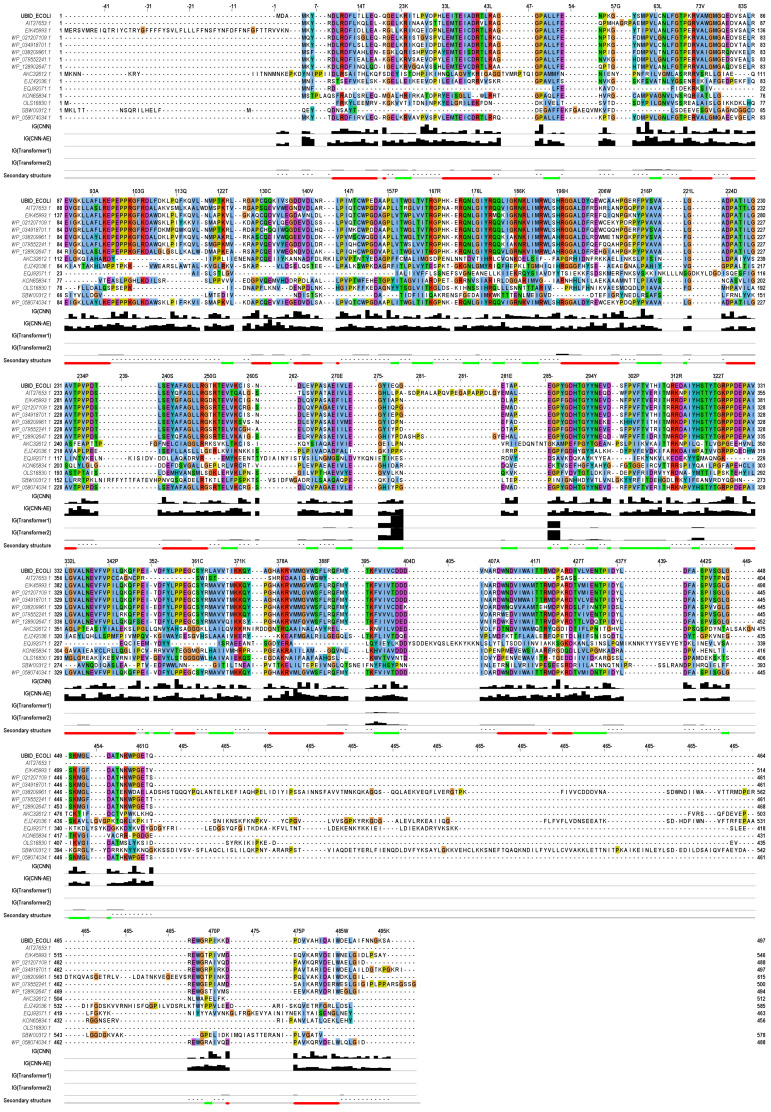
Multiple sequence alignment for E. coli UbiD (UBID_ECOLI) and representative UbiD sequences (Appendix A) and IG results of *E. coli* UbiD derived from each deep learning model. The secondary structures of *E. coli* UbiD registered as 2IDB [45] in the Protein Data bank [46] are shown below the alignment, helix, and sheet structures are displayed as red tubes and green arrows, respectively. Bar charts show the IG values that are normalized between 0 and 1. Transformers 1 and 2 represent IG values derived from feature vectors and prediction scores, respectively.

**Table 1 biology-12-00795-t001:** UbiD datasets for CNN, CNN-AE, and Transformer models.

Dataset Category	Training	Validation	Test
Positive data	1593	646	645
Negative data	62,476	8168	8167

## Data Availability

The data and source codes in this study are freely available at https://drive.google.com/drive/folders/1c_O5FDqXylDLx55e9duj-GbBPedPhWmA (accessed on 28 May 2023). The Python3 source codes can be utilized to build and evaluate 3 deep learning models, analyze models and extract protein features using IG.

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
