# Peer review of "Different Recognition of Protein Features Depending on Deep Learning Models: A Case Study of Aromatic Decarboxylase UbiD"

_biology, 2023, doi:10.3390/biology12060795_

Round 1
Reviewer 1 Report
The paper aims to explore important features of amino acid sites using Integrated Gradients and deep learning models to predict and extract features of UbiD enzymes. The paper concludes that each deep learning model understands protein features with different aspects from existing knowledge and has the potential to discover new features or functions of the protein. The approach is interesting and meaningful for discovering new important sites of the given protein. The study design is well described (even though it would be somewhat hard to follow for general readers in the Biology journal). However, it would be nice to address the following questions to help readers understand the research's context and significance.
- It would be helpful to provide a bit more background information about the problem that the paper is trying to solve. The positive data are UbiD sequences, and the negative data are non-similar sequences to UbiD. So, is the goal to detect the sequence features that distinguish the specific (UbiD) proteins from other sequences?
- The number of explained amino acid residues that are identified from their models is quite small. There is a need to explain why these sites are significant. Which amino acid residues are already annotated as functional sites? This would give readers a better sense of what the paper has achieved and what the implications are for future research.
- What about the correlation between each site's IG scores and sequence conservation?
- The references about the UbiD protein itself are pretty missing. The functionally important sites of UbiD, already reported in previous research papers, should be mentioned together with the prediction results and cited.
- The authors noted that deep learning models analyzed protein feature vectors to explore important features of amino acid sites. In this context, the definition of "importance" is somewhat ambiguous. Can the definition of the importance of amino acid location depend on the model and dataset used? The importance of the amino acid residues identified should be interpreted in more detail.
- The resolution of Figure 2 should be improved. All letters are hard to read.
As a non-native English speaker, I find it challenging to be certain about my language proficiency. However, I did not encounter any significant linguistic difficulties while reading the text.
Reviewer 2 Report
The manuscript analyzes the protein feature vectors generated by 3 deep learning models using Integrated Gradients to explore important features of amino acid sites. As a case study, prediction and feature extraction models for UbiD enzymes were built using these models. I have the following suggestions and comments:
1. When mention the deep learning methods for protein functions and structures prediction, several other widely-used methods should be included in Introduction: RaptorX (X Jing and J Xu, Nature Computational Science 1 (7): 462-469, 2022) and DMPfold (Greener JG et al, Nature communications, 10(1):3977, 2019) for protein structure prediction, I-TASSER-MTD (Nature Protocols, 17, 2326-2353, 2022) for protein structure and function prediction, and DEMO2 (Zhou et al, Nucleic Acids Research 50 (W1): W454-W464, 2022) for multi-domain protein structure prediction.
2. The reason for limiting sequence length to 400-700 should be discussed in 2.1.1.
3. How to define the Positive and Negative data should be described.
4. Figure 2 is important for the result analysis, but it is not clear in the current form, especially the words cannot be distinguished.
Reviewer 3 Report
This paper is dealing with an important topic of different recognition of protein features and a case study. Accurate computational model is highly desired, and this paper presents an approach based on CNN model, CNN-AE model and Transformer model. This paper was not sufficiently clear, and the whole problem-solving protocol was not precisely described and explained with the help of Figure. The amount of experiments in the whole article is also not adequate. The specific questions are as follows:
(1)The ablation experiments in this paper are relatively few, so it is suggested to supplement the experiments.
(2)There are traces of mutual copying and duplication in many parts of the article, please pay attention to correct in time.
(3)Additional methodological details for analysis, including programming code, should be included to facilitate reader research.
(4)The conclusion should supplement the innovation of the paper.
(5)The cited literature is not sufficient, and proper addition of relevant literature will make the article more convincing, such as 10.1038/s41586-022-04654-9, 10.1093/bib/bbac498, 10.1093/bib/bbac083. In addition, references are not standard, missing page numbers and volumes. Please do corrections.
(6)There are numerous grammatical and lexical errors in the article, which should be corrected so the reader can understand the intention accurately.
N/A
Round 2
Reviewer 3 Report
N/A
N/A